# Optical Rail Surface Crack Detection Method Based on Semantic Segmentation Replacement for Magnetic Particle Inspection

**DOI:** 10.3390/s22218214

**Published:** 2022-10-26

**Authors:** Lei Kou, Mykola Sysyn, Szabolcs Fischer, Jianxing Liu, Olga Nabochenko

**Affiliations:** 1Institute of Railway Systems and Public Transport, TU-Dresden, 01069 Dresden, Germany; 2Department of Transport Infrastructure and Water Resources Engineering, Faculty of Architecture, Civil- and Transport Engineering, Széchenyi István University, H-9026 Győr, Hungary

**Keywords:** crack detection, neural convolution, deep learning, semantic segmentation, rail surface

## Abstract

Railway damage detection is of great significance in ensuring railway safety. The cracks on the rail surface play a key role in studying the formation and development process of rail damage, predicting the occurrence of rail defects, and then improving the service life of the rail. However, due to the small shape of the cracks, the typical detection method is relatively complicated, and the speed is quite slow. Although traditional magnetic particle inspection technology is fairly accurate at detection, it is costly and inconvenient to carry and install, while also limiting the detection speed and affecting the system’s operation. In this paper, a semantic segmentation detection method is developed by using various collected rail surface crack data and deep learning through a neural network. By comparing the inspection of the same rail surface with magnetic particle inspection technology, only inexpensive cameras are used and the inspection speed is increased while maintaining relatively high accuracy. In addition, the method can achieve fast detection speeds if it is extended to be combined with high-frequency cameras. It is an economical, efficient, and environmentally friendly method for future rail surface detection.

## 1. Introduction

With the rapid global development of high-speed railways, railways have come to play a pivotal role in transportation, so the safety of rail transportation is directly related to the development of society and the economy. Maintenance spending by Deutsche Bahn in 2020 rose to EUR 1937.2 Mio. The maintenance cost of the upper equipment is 55% of the total cost. Most of this is due to the failure of the rails. Rolling contact fatigue (RCF) defects account for about 60 percent of defects found in data from Japan Railways. Therefore, checking the RCF defects in the course of routine railway maintenance is necessary. Rail cracks can easily expand downwards and are challenging to detect. If they are not detected and prevented in time, the rails will not be able to withstand high external forces. Ultimately, this can lead to safety incidents such as rail breakage. Therefore, the timely detection of rail cracks has become the most critical step in railway safety operations. Usually, the inspection method is mainly mechanical, resulting in severe waste of human resources and presenting problems with respect to timing and accuracy. If the operating costs are to be decreased, and the economic efficiency of transportation and traffic safety are to be increased, a transition to new technological equipment for diagnosing the technical condition of rails during technical maintenance and repair will be required [1]. Electromagnetic inspection, in this context, mainly refers to magnetic particle inspection (MPI), which localizes primarily surface and sub-surface defects. The electromagnetic equipment is not only large, but also expensive, therefore making it unsuitable for high-speed rail inspection vehicles. Therefore, the automatic detection of rail-surface damage replaces the approach that has been on the agenda. At the same time, the detection of surface cracks has an essential impact on the study of crack propagation modes and the further prediction of rail damage.

### 1.1. Existing Technology

Manual inspection is the primary method of rail-surface defect detection. However, it is difficult to obtain objective test results and defect locations by means of manual inspection due to subjective judgment. This detection method has the characteristics of subjectivity, low efficiency, and lack of convenience, and manual detection is not able to meet actual needs [2]. As early as 1928, the American SPERRY company used magnetic induction [3] to detect rail cracks, making the US the first country to use railway detection technology. With the in-depth research into deep learning in rail-crack detection technology, its application in fault detection has diversified. At present, non-destructive testing technology is widely used in railways, and the most commonly applied testing technologies include ultrasonic testing [4], eddy current testing [5], appearance testing [6], etc. Milosevic et al. (2022) [7] presented the simultaneous measurement of sleeper acceleration and six intersecting geometries on the basis of a scan. Their article developed a multibody simulation (MBS) model in combination with a structural rail model, and implemented swept intersection geometry to derive the link between the intersection geometry conditions and the resulting orbital excitations. Finally, Milosevic et al. (2022) realized the value of this technology for condition monitoring purposes. Kazemian et al. (2022) [8] studied wheel damage with the aid of finite element (FE) modeling. The researchers analyzed different types of rail damage by classifying vibrations. This method was applied to machine vision technology, and was able to eliminate the disadvantages of the subjective factors of manual detection. An acquisition speed of up to 60 km/h in orbit detection has become the research focus in recent years [9]. Intelligent vision inspection systems must capture track surface images through the use of an installed camera. Images are transferred to a computer, and the computer image-processing algorithm is used to detect defects and automatically complete the classification of defects.

Table 1 presents a comparison of the results described above, produced on the basis of an evaluation by the German non-destructive testing research company Foerste GmbH based on long-term testing practice [10]. Table 1 shows that both eddy current and magnetic particle inspections achieve a rating of A for the detection of metal surface defects. The eddy current testing requirements are higher. On the one hand, a set of eddy current testing equipment for rail surface defect detection requires at least several hundred thousand ohms for generally conductive components; on the other, rotational symmetry is required in the test objects. Rotational symmetry is necessary in order to perform the test as efficiently as possible. The component is rotated for testing and probing. This paper focuses on a rail surface detection method that is based on machine vision. Compared with magnetic particle inspection, it is simple to operate and performs detection faster, while still maintaining a high level of accuracy. 

Moreover, magnetic particle inspection can also achieve excellent detection results. This paper focuses on a method of rail surface detection that is based on machine vision. For the above reasons, we chose to perform a comparison with magnetic particle inspection as to whether it is able to detect large-area defects on the surface while at the same time achieving a similar effect to magnetic particle inspection in crack detection. Furthermore, it is not only simple to operate, but performs detection faster, and the cost is negligible compared to the above methods. While maintaining low cost, the company is able to perform multiple inspections in order to achieve the aim of the timely detection of the danger of failure. 

### 1.2. Working Outline

Traditional detection methods require strictly trained testing personnel. Otherwise, the accuracy of the test results will be seriously affected. Traditional detection methods are also inefficient due to the expense of technical equipment and the unintuitive nature of the results. Therefore, it is necessary to develop a fast and cheap detection method that meets current high-intensity detection needs. Computer vision detection is more intelligent, and possesses the advantages of rapid detection speed and intuitiveness of results, but the accuracy of this method also presents a challenge, and is affected by many factors. This leads to low detection accuracy. Cracks are thin, linear scars on the surface of a rail. With increasing rail service time, wear increases, and cracks on the rail surface will continue to evolve [11].

Because of the above challenges, the current paper proposes a new method to check for rail surface cracks, which can be further extended to the detection of other defects in the future in order to ensure the accuracy of crack identification. The main contributions are summarized as follows:

It is challenging to ensure crack detection accuracy under all environmental conditions due to the diversity and uneven distribution of cracks on the rail surface. First, three hundred and eighty data points were collected. These were obtained under different conditions and accumulated traffic volumes, in order to make the samples more balanced, increase the model’s generalized performance, and improve the detection accuracy with respect to various cracks.

On the basis of a comparison of the training of multiple neural networks, the most accurate algorithm for the detection of complex rail surface cracks was found to be that based on semantic segmentation. The semantic segmentation network based on DeepLabv3+, with ResNet has a high accuracy of 96.4% for rail surface detection.

For data with poor image quality, an image optimization method based on bilateral filtering was used to provide more acceptable crack detection as part of the crack recognition process in processed images following training. This improves detection accuracy while including as much crack information as possible.

The current paper proposes a crack connection method. The influence of other factors results in crack breaks being discontinuous and fine, resulting in inaccurate detection results. Our method connects the ends of fracture cracks—through an optimized crack propagation algorithm—in order to improve the detection accuracy.

On this basis, the crack detection of the same rail at the same position is realized on the basis of a full scan. In addition, the crack detection results are compared with those achieved using magnetic particle detection technology, acknowledging the possibility that cheap and fast machine vision technology is able to replace the vast and expensive particle imaging technology in crack detection.

## 2. Related Research and Methods

Crack detection is a typical linear target detection process from the perspective of vision, so the enhancement and positioning of crack images is a part of the research field of linear target detection. Compared with general linear targets, cracks have their own unique characteristics, the target width is relatively small, and the image contrast is low. Cracks exhibit discontinuity, bifurcation, miscellaneous points, etc., and they only show linear features in terms of the overall vision. Machine vision methods include traditional detection methodologies and deep learning detection techniques. Traditional automatic crack detection algorithms include threshold segmentation, edge detection, wavelet transformation, etc. These methods often assume that cracks have high contrast and good continuity throughout the whole image; however, this assumption is often invalid in practical engineering projects. Deep learning detection methods are able to learn rules from large amounts of data, on the basis of which algorithms can be used to accurately identify new samples. Their generalization ability, robustness, and detection effect are superior to those of traditional processing detection methods.

### 2.1. Traditional Image Processing Detection

Traditional detection methods must add some prepossessing steps before performing crack image processing. Image preprocessing is generally applied in image recognition, image representation, and other fields. In the process of image acquisition and transmission, the image quality is often reduced for various reasons. It is usually difficult to detect and extract targets when analyzing crack images due to interference from noise and other problems. For instance, if the object’s shape and size are too great, the image features may not be easily identifiable. In addition, due to the nature of the process, some distortions and deformation may occur in the image. The abovementioned factors can affect the feasibility of processing the image. Machine vision detection mainly focuses on improving image quality by performing various computational steps to transform the image into a surface that highlights certain pieces of interesting information. These steps include image contrast enhancement [12,13,14], image filtering [15,16], and edge extraction [17,18]. When it comes to analyzing crack images, it is usually difficult to detect and extract the targets due to the presence of interference from noise and other problems. In addition, due to the nature of the process, passenger comfort issues may also be experienced. Traditional image processing methods mainly focus on detecting large defects and obvious long cracks. 

### 2.2. Deep Learning Detection

In recent years, researchers have proposed many methods for automatically detecting surface defects, among which the data-driven deep learning method has attracted a great deal of attention. Deep learning is a machine learning method derived from the study of artificial neural networks that mimics the human thinking mechanism to process and analyze data. Trinh et al. [19], at the Washington Research Center, proposed a real-time automatic vision track detection system for important rail components such as tie bars, tether plates, and anchor bolts. The method has high accuracy and efficiency, but can only detect anchor bolts and other rail components and has poor expandability. Gibert et al. [20] proposed a method for analyzing images and detecting track surface defects using a deep convolutional neural network. In [9], rail surface detection methods based on deep learning were surveyed, analyzed, and compared in detail. The speed of machine vision detection is faster than that of physical detection, so the application of machine vision in rail defect detection systems for the realization of high-speed rail surface crack detection may become a significant trend in the future. However, there are few detection methods for rail surface cracks. Although classical target detection methods have achieved relatively good results, the background of the crack detection image is more complex. Additionally, crack density is high in some areas, there may be interference from image noise, and the generalization and accuracy of the detection methods are low. With the continuous development of deep learning technology, semantic segmentation algorithms based on deep learning have been widely used in image classification, object detection, image segmentation, and other applications. In this paper, the semantic segmentation method is used to detect rail surface cracks.

## 3. Proposed Solution

This chapter introduces the standard methods for performing accurate crack detection, including establishing the neural network, image data acquisition, image optimization, and crack connection. The efficient combination of these methods makes it possible to realize the accurate detection and identification of rail surface cracks.

### 3.1. The Network Structure

Semantic segmentation is a method for performing classification at the pixel level. All pixels belonging to the same category should be classified into one category. Therefore, semantic segmentation can be used to understand images at the pixel level. Deep learning has achieved great success in the field of semantic segmentation. In 2014, a fully convolutional network (FCN) was developed. FCN replaces the network’s complete connection layer with convolution, thus making it possible to input an image of any size much more rapidly than by using the Patch Classification method [21]. The currently available deep learning semantic segmentation models were developed based on FCN. Deeplab V2, proposed in 2016, uses empty convolution [22] to increase the receptive field without increasing the number of parameters. Using the methods developed in the present paper, it is possible to improve the segmentation network. In 2017, the team improved the pyramid void pooling method in the spatial dimension [23], as shown in Figure 1. Meanwhile, the ResNet model was enhanced by using void convolution/multi-space convolution. To achieve further improvement, two types of neural network were used in DeepLabv3+, a spatial pyramid module and an “encoder–decoder structure”, to perform semantic segmentation [24].

This paper uses the network to perform image segmentation in order to obtain rail crack images. With the deepending of the network, the accuracy of the training set decreases [25]. The authors can confirm that this is not caused by overfitting (in the case of overfitting, the accuracy of the training set should be high). Overfitting is particularly likely to occur when there is a small amount of data, so we take the residual network as one of the leading training network structures in this paper. Therefore, the authors propose a new kind of network for this problem, called a deep residual network, Residual Net, where the data output of a specific layer in the front layer is introduced to the input part of the data layer in the back layer by skipping the layers. This means that one of the preceding layers will contribute linearly to the content of the subsequent feature layer. This allows the network to be as deep as possible, introducing a completely new structure, as shown in Figure 2. The residual is used to design and solve the degradation problem. It also solves the gradient problem and improves the performance of the network. The left side of Figure 2 shows the residual network structure with different layers. On the right is the residual block, the key structure of the residual network. The residual block can be understood to be adding some shortcut connections to the forward network. These connections skip certain layers and pass raw data directly to the following layers. The newly added shortcut connection will not increase the number of parameters or complexity of the model.

ResNet50 has two basic blocks, named Conv-Block and Identity Block, respectively. The input and output dimensions of the Conv-Block are different, so they cannot be connected in series. Its function is to change the dimensions of the network. For the Identity Block, the dimensions of the input are the same as the dimensions of the output, and they can be connected in series to deepen the network. In this paper, ResNet50 was used as the backbone for segmentation tasks, and the deep lab v3+ neural network structure composed of Atrous Spatial Pyramid Pooling (ASPP) and aux-classifier is added. The network input data input (H, W) are the same as the network output (H, W), significantly reducing the insufficiency of data information caused by the limitation regarding the number of images. The model’s performance on semantic segmentation tasks in the whole network is further improved by down-sampling and up-sampling. Figure 3 shows the basic structure of the neural network combining Deeplabv3 and ResNet 50.

### 3.2. Data Preparation

#### 3.2.1. Magnetic Particle Inspection

To verify the accuracy of the semantic segmentation data, magnetic particle inspection (MPI) was used to detect the characteristics of frog track surfaces with high crack density from the beginning to complete damage. However, this method is time consuming, less able to be automated, and requires expensive and bulky equipment. At the same time, a standard camera was used to collect information on the rail surface in the same state. Online magnetic particle flaw detection speed is up to 0.75–1.0 m/s. The magnetic particle flaw detector first uses the magnetic pole to magnetize the rail and then sprays fluorescent magnetic powder onto the surface of the entire rail with a spray gun. The magnetic powder is attracted to the surface defects, and manual or computer-controlled sensors are then able to identify defective rails. After comparison with the manually processed sample rails according to the standard requirements, only those rails with quality control defects exceeding the standard were marked, in order to provide a basis for inspectors’ grading and grinding. The magnetic particle flaw detector was able to react sensitively to cracks with depths of over 0.13 mm, 0.25 mm, and 0.38 mm, as well as other defects. Its advantages include less required investment in equipment and high reliability. The disadvantage is that the operation cost is high, and the defects cannot be accurately classified. Figure 4 shows MPI and camera images of fatigue damage on the rolling surface of a frog track during regular operation with 33 Mt accumulated traffic. The reason for choosing this period is that the number of cracks at this time is complex, and the accuracy of the semantic segmentation detection method compared to the alternative magnetic particle detection can better be evaluated. The MPI images clearly show the image patterns of cracks in different areas on the frog track surface. Ref. [26] regarded cracks in camera images to be relatively less clear than those in MPI images. However, experienced experts are able to detect cracks, and believe that the information content in camera images is no lower than that in MPI images. The main problem with high-resolution photodetection is that automatically identifying cracks in their early stages is challenging.

#### 3.2.2 Training Image Preparation

In recent years, feature learning using deep neural networks has been applied to various computer vision and classification problems, and has been demonstrated to be successful in many fields. Multiple benchmarks can be used to describe the classification accuracy of visual data. Usually, each has a large number of samples. To achieve accurate crack image segmentation, our team, over the course of a year, collected 380 images determined to contain crack damage from more than 20 rails under different weather conditions, different accumulated traffic volumes, and a variety of wear situations. As shown in Figure 5, the set of images in the collection has a complex variety of lighting conditions. The cracks are of various types, including grain structures on all rail surfaces. To improve the resistance of the convolutional network to environmental impacts, the network was only classified before training, and no other image preprocessing operations were performed. The labeled images were divided into three three categories: rail surface, rail crack, and rail damage. All fine cracks are marked during the labeling process, although manual inspection of these cracks requires careful magnification in order to ensure that the trained neural network had as comprehensive a crack recognition ability as possible.

### 3.3 Image Optimization

There are not many methods of color image optimization. In this paper, the popular automatic color channel enhancement is not adopted for color image optimization [27], not are the basic histogram methods. Rather, we adopt the bilateral filtering method, which preserves the basic characteristics of the pixel region while enhancing the image. This method is more suitable for semantic image segmentation optimization. *Bilateral filtering* is a nonlinear method combining the image’s spatial proximity and pixel value similarity. It considers the spatial information and gray level similarity simultaneously to achieve the purpose of edge preservation while removing noise. An ordinary Gaussian filter would blur the edges of the image, while a bilateral filter has the property of edge preservation [28,29]. Before introducing relevant formulas, a point (*i*, *j*) is defined as the coordinates of the central point. Point (*k*, *l*) is any point in the adjacent area centered on point (*i*, *j*). The spatial distance from point (*k*, *l*) to point (*i*, *j*) in the spatial matrix is defined as:(1)$$d\left(i,j,k,l\right)=\exp(-\frac{{\left(i-k\right)}^{2}+{\left(j-l\right)}^{2}}{2\sigma_{d}^{2}}\;)$$

It is worth noting that once $\sigma_{d}^{\;}$ has been defined, the values of each point in the spatial matrix remain constant. Moreover, the range matrix can be defined as:(2)$$r\left(i,j,k,l\right)=\exp\left(-\frac{∥f\left(k,l\right)-f\left(i,j\right){∥}^{2}}{2\sigma_{r}^{2}}\;\right)$$

Here, $\sigma_{d}^{\;}$ is the global variance and $\sigma_{r}^{\;}$ is the local variance. After multiplying the two, a data-dependent bilateral filtering weight matrix *w* (*i*, *j*, *k*, *l*) is produced:(3)$$w\left(i,j,k,l\right)=\exp(-\frac{{\left(i-k\right)}^{2}+{\left(j-l\right)}^{2}}{2\sigma_{d}^{2}}-\frac{∥f\left(k,l\right)-f\left(i,j\right){∥}^{2}}{2\sigma_{r}^{2}})$$

Finally, the new pixel value g (*i*, *j*) of point (*i*, *j*) is obtained.
(4)$$g\left(i,j\right)=\frac{\sum_{\left(k,l\right)\in S}\;f\left(i,j\right)w\left(i,j,k,l\right)}{\sum_{\left(k,l\right)\in S}\;w\left(i,j,k,l\right)}$$

The function selects the weight according to the distance of the pixels. The closer the distance is, the greater the weight will be. It is the same as box filtering and Gaussian filtering. The R function assigns weights according to differences in pixel values. In the flat region, the pixel difference is slight, and the corresponding weight of the range *r* (*i*, *j*, *k*, *l*) is close to 1. At this time, the spatial weight d (*i*, *j*, *k*, *l*) plays a significant role, which is equivalent to Gaussian blur directly on the flat region.

It should be noted that the spatial kernel is always constant, while the range kernel varies from region to region. The edge possesses a characteristic whereby the pixel values of points close to each other differ significantly. Therefore, it is desirable to preserve this edge property while filtering. Therefore, because the difference between pixels is prominent in the edge region, the difference between *f*(*k*, *l*)–*f*(*i*, *j*) is enormous. Then, the kernel weight of the value domain becomes smaller. This leads to a decrease in the total weight value *w* (*i*, *j*, *k*, *l*) (*w*= *r* × *d*), and the current pixel (*i*, *j*) is less affected by this great difference, thus maintaining the details of the edge. Figure 6 shows the original image, the cracks detected before optimization, and the comparison after optimization. The optimization effect of this method is noticeable, and more surface crack information can be obtained.

### 3.4 Direction-Based Crack Connection

In crack detection, the depth of the crack itself is not uniform at each position, the effect of lighting and image processing is different, and the same crack can be identified as several intermittent sections. At this point, it may result in misjudgement to qualify the rail surface by counting the length and number of cracks on the rail surface. In the experience-based fatigue evaluation of rails, workers mainly count the length and number of cracks. Conversely, deep neural networks can be used to extract the crack characteristics. It is necessary to connect these disconnected cracks using the region growth method. The traditional region growth method uses seed pixels in around four or eight neighborhoods within the region, and determines whether or not to group neighborhood pixels within a seed collection point based on a comparison of grayscale, texture, or color features. New seed point are successively formed, and eventually, all suitable pixels will grow into a connected area. However, experiments have shown that the traditional region growth algorithm [30,31], which takes pixels as units, has no obvious efficacy when dealing with broken cracks.

On the one hand, the same crack where there is a gap of a considerable distance will still not be connected. On the other hand, different cracks that are close to one another will be mistakenly connected into the same crack. The shape of a crack is that of a long strip, and possesses the characteristic of extending along a direction. Therefore, an improved region growth algorithm based on direction was proposed. The crack region is used as the seed point, the crack direction and the distance between the crack endpoints in different regions are used as the growth conditions, and better crack connectivity is achieved. A diagram of the growth principle of the crack area is shown in Figure 7. Before using neural network identification and determining the crack location and size of images, each time through the loop, only a certain crack direction is selected. Then, cracks in this picture c are eliminated, the center of the crack radius R1 and R2 is selected, the angle is selected with the crack itself as the positive and negative theta, and the range can be framed within the blue shape in Figure 7. The four points are B1, B2, B3, and B4. 

The calculation formulas are:(5)$$\begin{array}{c} x_{B1}=x_{o}+R_{1}\times cos\left(\alpha +\theta \right),y_{B1}=x_{o}-R_{1}\times sin\left(\alpha +\theta \right);\\ x_{B2}=x_{o}+R_{1}\times cos\left(\alpha -\theta \right),y_{B2}=x_{o}-R_{1}\times sin\left(\alpha -\theta \right);\\ x_{B3}=x_{o}+R_{2}\times cos\left(\alpha +\theta \right),y_{B3}=x_{o}-R_{2}\times sin\left(\alpha +\theta \right);\\ x_{B4}=x_{o}+R_{2}\times cos\left(\alpha -\theta \right),y_{B4}=x_{o}-R_{2}\times sin\left(\alpha -\theta \right). \end{array}$$

In this paper, *R*_1_ and *R*_2_ are the two thresholds of the crack radius, respectively. According to the actual distribution of cracks on the surface of the parts, *R*_1_ = 4, *R*_2_ = 10. P_1_ can be scanned in the box enclosed by these four points. Then, P_1_ is connected with P_2_. Considering that the distance between the ends of the same crack is relatively small, the distance between the ends of cracks in different regions can be used as another growth condition in the region growth algorithm. For any two broken crack regions, the distance between their endpoints will be:(6)$$s=\sqrt{{\left(i_{2}-i_{1}\right)}^{2}+{\left(j_{2}-j_{1}\right)}^{2}}$$
where (*i_1_*, *j_1_*) and (*i_2_*, *j_2_*) are the coordinates of the endpoints of the two crack regions in the image plane. When the distance S between the endpoints of the two crack regions is in the blue box and below the threshold limit, the crack region is considered to meet the zone growth condition and will be added to the new seed point. For region growth, P_1_(*x*_1_,*y*_1_) and P_2_(*x*_2_,*y*_2_) are the starting point and ending point, respectively, and the position of the point P(*x*_i_, *y*_i_) between the two points is:(7)$$k=-\frac{\left|y_{1}-y_{2}\right|}{\left|x_{1}-x_{2}\right|}$$
(8)$$\;\;x_{i}=x_{i-1}+1;y_{i}=y_{i-1}+k.$$

## 4. Results

In this section, the semantic segmentation results are compared and analyzed. Then, the differences between the segmented image and the actual photo after scanning the whole larger image are compared. Then, we analyze whether region growth is an effective method, and finally the differences and advantages of crack recognition technology based on semantic segmentation with respect to magnetic particle inspection are evaluated. Finally, it is determined whether the present method can serve as a replacement for magnetic particle inspection.

### 4.1. Semantic Segmentation Results

In most studies of rail surface crack detection, the calculation formula for crack defects is used to find the proportion of correctly identified cracks to the total number of cracks detected. First, it is necessary to recall how many actual cracks were identified. In this paper, 80% of the data were used for training, and 20% of the data were used for validation. The number of cracks is corrected to the actual number of cracks by calculating the validation accuracy. It is well known that convolutional neural network (CNN) models can be used to create this prediction concept, in which images can be divided into training and validation datasets containing both faults and fault-free images. After repeated training of the model, correct predictions can be obtained for the images, including whether there is a fault present, and thus it is able to return those images with defects and those without [32]. In order to be able to replace the results obtained using magnetic particle inspection, the method in this paper should not only be able to identify the presence of cracks, but also be able to identify fine cracks to the greatest extent possible. It is committed to including all cracks that can be detected by magnetic particle inspection. Based on Deeplabv3+ and ResNet50, U-net [33], SegNet [34], Xception, and Inceptionresnetv2—which have been prevalent in recent years—are also segmented using DeeplabV3+, and the results are compared. Figure 8 shows the variation in the accuracy of the five algorithms as a function of number of iterations. The apparent red curve has less undulation and consistently high accuracy. In terms of validation accuracy with the number of training iterations, ResNet50 achieved the best result of 96.4%. According to the experimental results, the algorithm in this paper is able to extract cracks well from complex backgrounds.

The comparison of the original image, the image label, and the training results (prediction) shown in Figure 9 shows that the results meet the requirements of basic flaw detection. When the crack is relatively straightforward and visible, the training results generated based on the database in this paper are able to accurately identify the crack. In contrast, when the crack is relatively small, or subject to strong light reflection, the training results are relatively poor, resulting in crack discontinuity and other results. Therefore, a reasonable crack growth method is essential to the method proposed in this paper. The cracks, defects, and backgrounds in the five images are accurately distinguished in the results shown in Figure 9. Only the crack in the fourth image produces a discontinuity. Therefore, we use it as an example to demonstrate the crack connection algorithm.

For the extracted defects, it is necessary for the traditional method to judge whether there is a defect by using the statistics of the defect area and aspect ratio. When the aspect ratio reaches a specific value, it can be judged that the defect at this location is a crack defect. However, the method in this paper is able to directly distinguish between cracking and spalling. Therefore, this step is omitted, which is also a great advantage of the method in this article.

### 4.2. Crack Connection Results

In the case of Figure 9, there are two pictures due to the limitations of the shooting conditions, which seriously affected the picture quality, and the cracks are (obviously) broken. Therefore, this example is taken their repair using the crack growth method. As shown in Figure 10, it can be seen that this method has a good effect on intermittent cracks and is able to effectively connect them. It is a perfect result. The crack growth algorithm used in this paper positively affects the connection of cracks under complex crack conditions. This has significant advantages compared to the method described in [35,36], which were only able to connect two or three cracks in a single image. This method can also be widely used in other crack detection fields.

### 4.3. Comparison with MPI

It can be seen from Figure 11 that the crack parameters detected by the algorithm proposed in this paper are in good agreement with the actual crack parameters. The length of the crack can be defined as the sum of the lengths of its connected domains. To verify the accuracy of the algorithm, crack detection was performed for the photo presented in Figure 4. The MPI crack images of the corresponding fragments were compared. Although the algorithm’s accuracy is 96.4%, this does not mean it is a suitable replacement for MPI for the purposes of statistical comparison. This paper defines the number and length of cracks detected in MPI as the reference base. In this section, we select four rail images on which to perform detection directly using the algorithm introduced in this paper. Furthermore, the detection results are compared with the MPI results obtained at the same locations, as shown in Figure 11.

It is assumed that the crack length corresponding to the crack detection in images was detected using MPI. Supposing that the method in this paper is able to detect 50% of the corresponding crack length obtained using MPI when performing crack detection on the image, it can therefore be considered that the crack has been detected. When 80% of the corresponding crack length obtained using MPI is detected when performing crack detection on the image, it can be considered that the crack has been accurately detected. To quantitatively evaluate the crack extraction performance on the basis of the above variables, the resulting detection rate *E_r_* is expressed as:(9)$$E_{r}=N_{r}/N$$

*N_r_* is the number of cracks for which more than 50% of the corresponding length registed using MPI can be detected. *N* is the number of cracks detected using the MPI inspection method. The accurate detection rate *E_A_* is expressed as:(10)$$E_{A}=N_{A}/N$$

*N_A_* is the number of cracks for which more than 80% of the corresponding length registered using MPI can be detected. *N* is the number of cracks detected using the MPI inspection method.

Comparing the values of *Er* and *E_A_*, the average value of *Er* is 88.03%, and that for *E_A_* is 82.91%. The method proposed in this paper is able to completely replace the MPI detection method. Figure 11 shows the results obtained using the authors’ method compared with those obtained using camera and MPI images. The four figures correspond to the results presented in Table 2. It is evident that our method is perfectly able to match the crack size and location performance of the other two methods. Nevertheless, because some natural conditions can affect the collected images, the detection some cracks is not accurate enough. The detection method presented in this paper has a more accurate detection rate for many cracks that MPI is not able to detect, such as in the edge area of rail surfaces with significant damage. In this respect, the method proposed in this paper has a higher accuracy for the detection of dangerous cracks. Therefore, it is better able to play a role in ensuring the safety of railway operations. Furthermore, the accuracy of this method can be improved further by equipping an artificial light source when performing crack detection.

It can be seen from the above two comparisons that the overall detection accuracy of the method proposed in this paper for surface defects is 96.4%, while the average overall accuracy of crack detection is 87.83%. We present a comparison the detection methods and accuracy of some other methods presented in the literature with our method in Table 3.

The comparison results in Table 3 demonstrate that the results presented in this paper are not inferior to the results obtained using other methods, and even present certain advantages. The main purpose of this paper is to find a more economical and efficient detection method. In terms of accuracy, our proposed method is able to achieve an effect similar to that of magnetic particle inspection.

## 5. Conclusions

In this paper, a new visual detection system for rail surface defects was developed, emphasizing the crack detection algorithm. The main contribution of this paper is the proposition of a crack detection method that is able to replace magnetic particle detection. More specifically, in extracting crack characteristics, we proposed the collection of images of rail surfaces according to the degree of complexity of the environment, under a wide range of different lighting conditions, natural environments, and degrees of damage. Moreover, a magnetic particle flaw detector was used to collect accurate crack information on the rail surface in order to verify the accuracy of this method. During the process of determining crack intensity under conditions in which there is less crack information available, the color images of the rail surface will automatically be enhanced using color level processing, so that a sufficient amount of information will be entered into the statistical characteristics. Then, a new crack growth algorithm is used to connect the crack fracture edges. More specifically, the short gaps between cracks are filled with spatial information in order to recover incomplete cracks, and the crack strike is used to determine the direction of scanning joints. Finally, the results of the recognition using the semantic segmentation method are compared with those obtained with the magnetic particle inspection method for the same area with respect to crack information on the rail surface to obtain an information image comparison. The experimental results show that the method presented in this paper encompasses crack information obtained by magnetic particle inspection while also being able to replace magnetic particle inspection. Moreover, this method only needs simple and portable shooting equipment, and the detection speed and equipment price present tremendous advantages over magnetic particle inspection.

Inexpensive and efficient detection methods are increasingly becoming able to meet detection needs. Under the same inspection cost, it is possible to perform an increasing number of inspections in order to ensure the safety of railway operations. However, while the proposed algorithm is able to detect cracks on the rail surface, there are still some problems that need further consideration. First, other crack classification methods based on a deep neural networks will be further studied to distinguish between different types of crack and to eliminate other interference factors. Second, graphics processing units or fast techniques will be considered in order to improve detection speed.

## Figures and Tables

**Figure 1 sensors-22-08214-f001:**
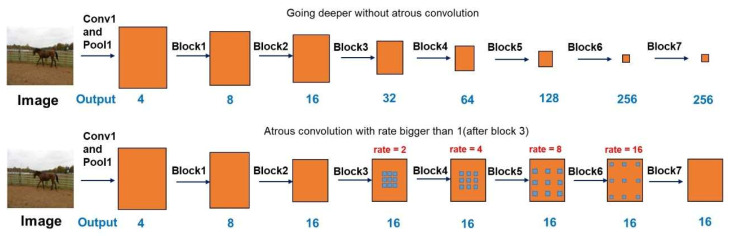
Cascaded modules without and with atrous convolution [23].

**Figure 2 sensors-22-08214-f002:**
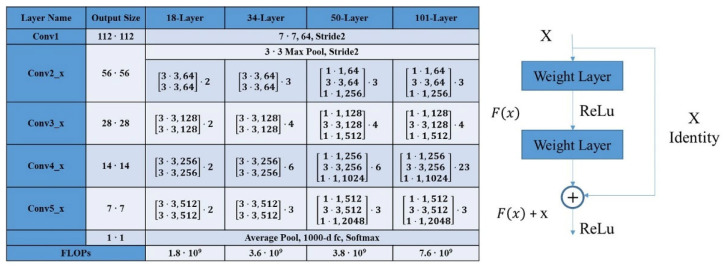
Architectures for ImageNet and Residual Learning: a building block.

**Figure 3 sensors-22-08214-f003:**
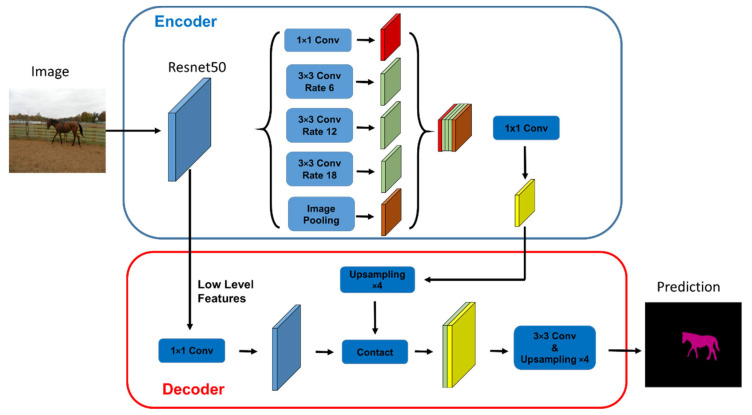
The network structure combining Deeplabv3 and ResNet 50.

**Figure 4 sensors-22-08214-f004:**
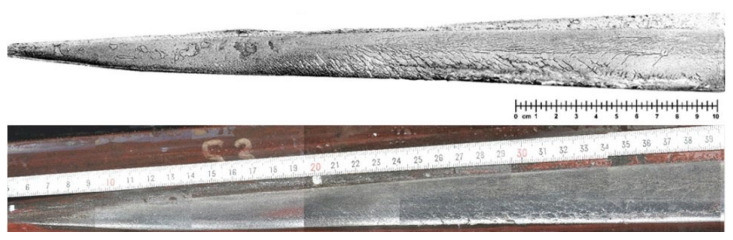
Magnetic particle inspection (above) and camera (below) images of the rolling surface on a frog nose after 33 Mt [26].

**Figure 5 sensors-22-08214-f005:**
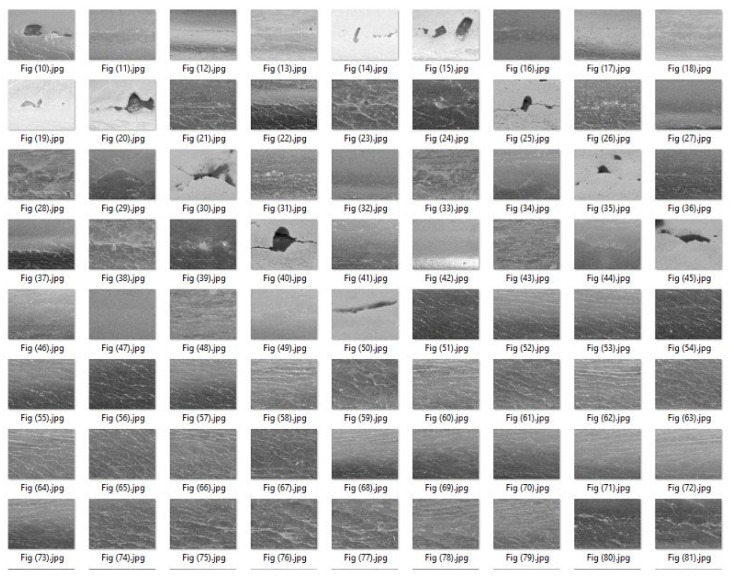
Images under different lighting conditions.

**Figure 6 sensors-22-08214-f006:**
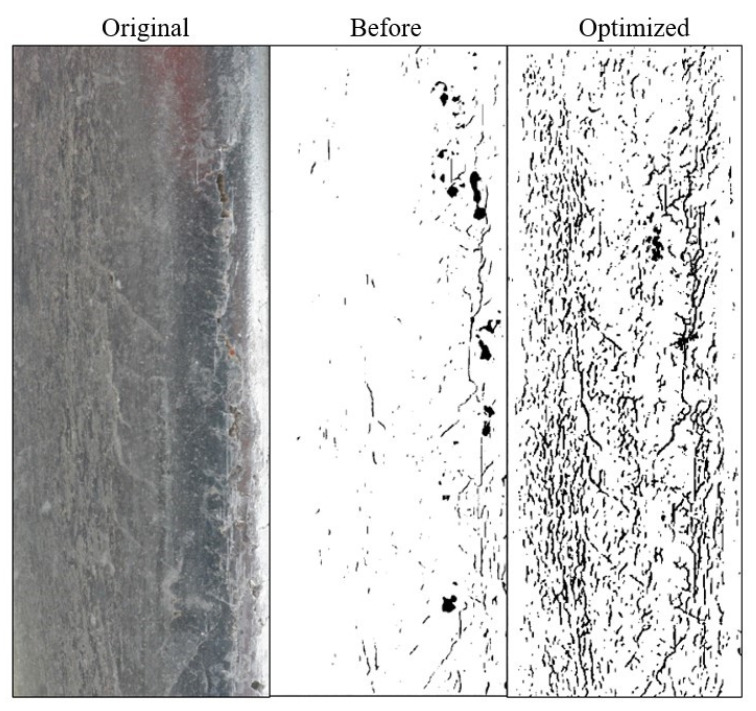
Optimized image.

**Figure 7 sensors-22-08214-f007:**
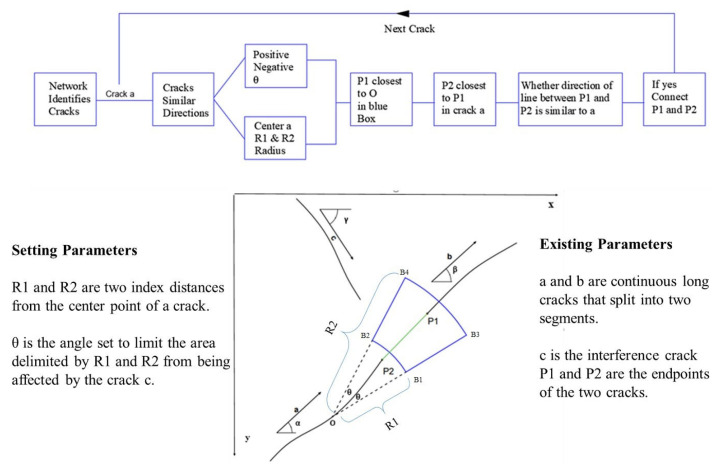
Schematic of scratch region growth.

**Figure 8 sensors-22-08214-f008:**
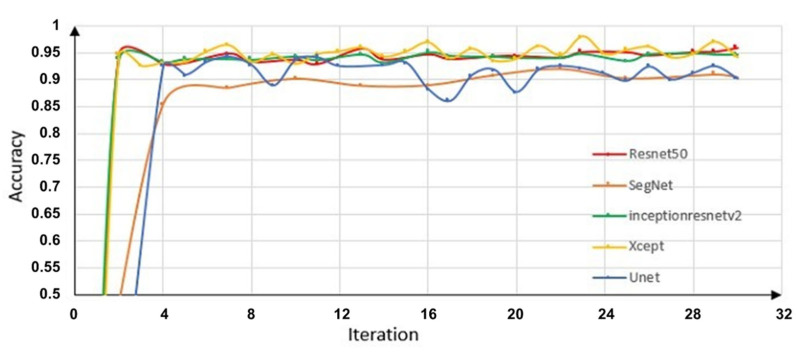
Comparison of the accuracy of the proposed network with that of other networks by number of iterations.

**Figure 9 sensors-22-08214-f009:**
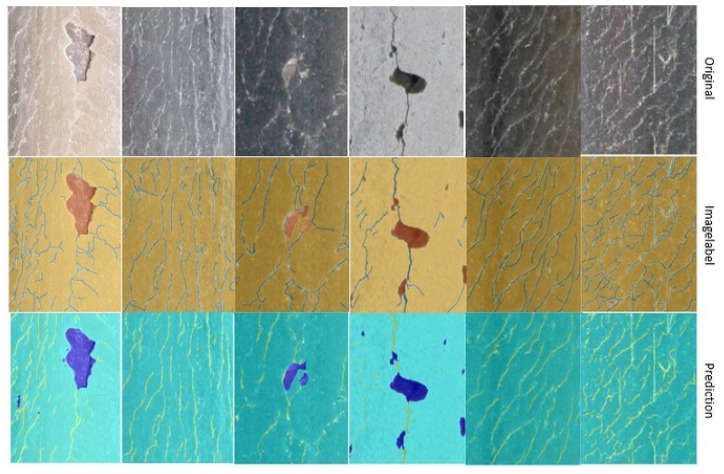
Semantic segmentation results.

**Figure 10 sensors-22-08214-f010:**
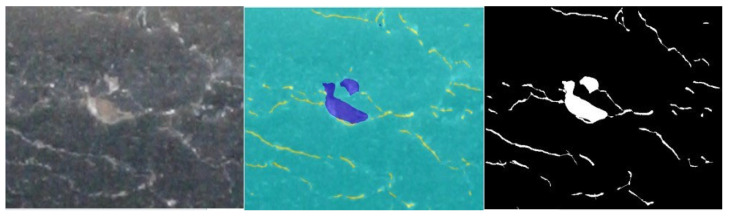
Cracks after connection.

**Figure 11 sensors-22-08214-f011:**
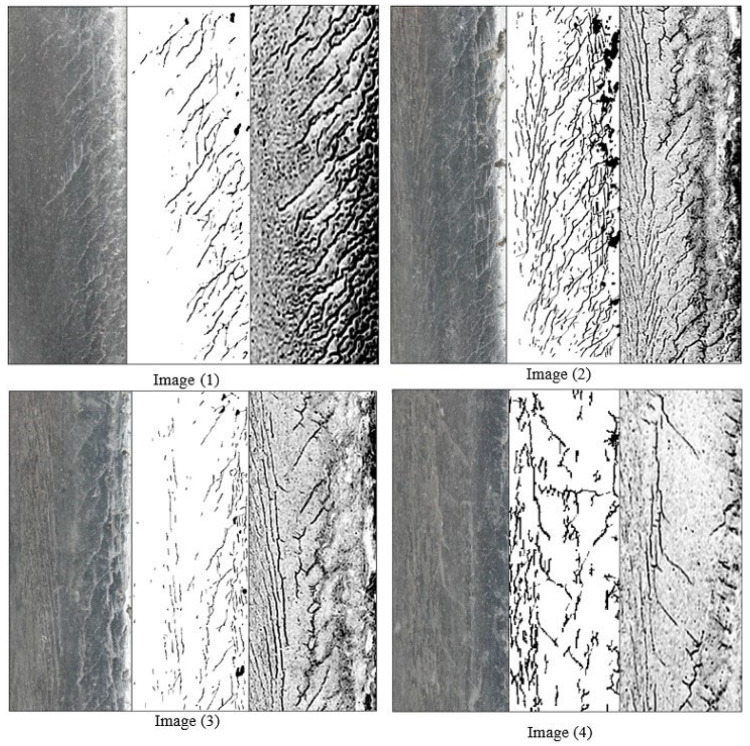
Comparison with MPI in four different sections of rail.

**Table 1 sensors-22-08214-t001:** Ratings of the various detection methods by Foerster GmbH.

Flaw Type		Eddy Current	X-ray	Magnetic Particle	Ultrasonic
	Method				
Surface Breaking, Linear		A	C	A	D
Surface Breaking, Linear & Parallel to Surface		A	D	A	D
Near-Surface		B	A	B	B
Subsurface, Volumetric		D	A	D	A

A = most suitable; B = kind of OK to use; C = not so good; D = not at all suitable.

**Table 2 sensors-22-08214-t002:** Accuracy comparison.

Image No.	MPI	Segmentation 50%	Segmentation 80%	E*_r_*	*E_A_*
Image 1	23	19	17	81.82%	73.91%
Image 2	34	32	30	91.18%	85.29%
Image 3	16	14	13	84.21%	81.25%
Image 4	17	16	15	94.12%	88.24%

**Table 3 sensors-22-08214-t003:** Comparison of results with other methods.

Method		Objtect	Accrancy
Ours	Semantic Segmentation	Defect	96.4%
		Cracks	87.83%
Article [37]	Bayesian classification	Defect	75%
Article [38]	YOLOv4	Defect	92.54%
Article [39]	k-Means Clustering	Defect	90.9%
Article [40]	Eddy Current	Crack Depth < 2 mm	86%
		Crack Depth: 2–6 mm	88%
Article [41]	Magnetic Particle Inspection	Crack Length: 2.5–5.0 mm	95%
		Crack Length: 2–2.5 mm	50%
		Crack Length: 1–1.5 mm	30%
		Crack Length < 1 mm	0%

## Data Availability

The datasets used and/or analyzed during the current study are available from the corresponding author upon reasonable request.

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
