# Peer review of "Optical Rail Surface Crack Detection Method Based on Semantic Segmentation Replacement for Magnetic Particle Inspection"

_sensors, 2022, doi:10.3390/s22218214_

Round 1
Reviewer 1 Report
Overall, the article is well-written and clearly articulated. I was able to find only the following, easy to fix, issues:
- Figure 3 is a slightly altered and unreferenced image, the source should be referenced. Also, at the sides of the image, parts of the cropped text is still visible.
- The image quality of most of the figures is quite low.
- In formula 6, a bracket is missing.
Author Response
- Figure 3 is a slightly altered and unreferenced image, the source should be referenced. Also, at the sides of the image, parts of the cropped text is still visible.
Response: We have redrawn Figure 3. Because the network structure framework is known and accepted. We just combined it with Resnet50.
- The image quality of most of the figures is quite low.
Response: We have replaced all images with higher quality images as much as possible.
- In formula 6, a bracket is missing.
Response: Thanks to the reviewers for their careful review, this error has been corrected. Added a bracket to the formula.
Reviewer 2 Report
Dear author:
The basic science of this paper is well developed with appropriate standards. The author and his team have developed a railway damage detection method based on semantic segmentation, which uses the collected data of various rail surface cracks to conduct in-depth learning through convolutional neural network. Compared with traditional techniques, this method maintains relatively high accuracy, uses only cheap cameras, and has faster detection speed. However, due to the complex site environment and unpredictable events, it is difficult to ensure the accuracy of this method. I hope my notes will be helpful to the revision of the manuscript.
Here are some things that need to be revised:
In the summary section, the author hopes to introduce the detection method proposed in this paper in more detail, and use the experimental results to illustrate the effectiveness of this method.
In the introduction, the author only mentions the economic significance of this study, and the significance to railway safety engineering after the method change is not explained.The paper should first start with a reference to it. In section 1.1, into the method proposed in this paper is compared with magnetic particle inspection. No comparison is made with ultrasonic inspection, eddy current inspection, and appearance inspection mentioned above. And there is a lack of detailed description and analysis of advantages and disadvantages of ultrasonic inspection, eddy current inspection, and appearance inspection.
In Part 3, I hope the author can briefly explain why he chose Restnet. In addition, I suggest that the author divide the data set into training set, test set and verification set. In the Figure 5 section, I suggest picking a few typical images and describing them. Clear pictures will make the reader's reading experience better. Numbers taken with cheap cameras may not impress readers.
In Part 4, I suggest we discuss the experimental data in more detail.
For the fifth part, only the method in this paper is compared with magnetic particle testing. It is not compared with other methods mentioned in this article. The advantages and disadvantages of this method can be compared with those of other methods mentioned above to explain in more detail why this method has better application prospects. According to the author's experiment, the accuracy is only 94.6%, which may lead to the failure to detect the cracks on the rail surface.
In addition, this article is a highly specialized article, in which some abbreviations are used. For ordinary readers, these abbreviations will make it difficult for them to read the article. I suggest providing explanations where these abbreviations first appear. Moreover, the writing of this paper needs a lot of improvement in grammar, spelling and expression. This paper needs careful English polish, because there are many typing mistakes and poorly written sentences.
I hope authors will improve this study and resubmit again in this journal.
Best Regards

Author Response
- In the summary section, the author hopes to introduce the detection method proposed in this paper in more detail, and use the experimental results to illustrate the effectiveness of this method.
Response: We added more detailed descriptive sentences to the images shown in the experimental results. In the end, it proves the effectiveness of this method by comparing it with other scholars' research results.
- In the introduction, the author only mentions the economic significance of this study, and the significance to railway safety engineering after the method change is not explained. The paper should first start with a reference to it. In section 1.1, into the method proposed in this paper is compared with magnetic particle inspection. No comparison is made with ultrasonic inspection, eddy current inspection, and appearance inspection mentioned above. And there is a lack of detailed description and analysis of advantages and disadvantages of ultrasonic inspection, eddy current inspection, and appearance inspection.
Response: We choose to compare with the results of magnetic particle inspection because this method is rated A for metal surfaces in the Foerste GmbH. We supplement the rating results of Foerste GmbH at the end of this subsection. The advantages of each method can be significant compared through the rating results.
|
Flaw Type Inspection Method |
Eddy Current |
X-Ray |
Magnetic Particle |
Ultrasonic |
|
|
Surface Breaking, Linear |
A |
C |
A |
D |
|
|
Surface Breaking, Linear & Parallel to Surface |
A |
D |
A |
D |
|
|
Near-Surface |
B |
A |
B |
B |
|
|
Subsurface, Volumetric |
D |
A |
D |
A |
|
|
A = Best suitable; B = Kind of OK to use; C = Not so good; D = Not at all suitable |
|||||
- In Part 3, I hope the author can briefly explain why he chose Restnet. In addition, I suggest that the author divide the data set into training set, test set and verification set. In the Figure 5 section, I suggest picking a few typical images and describing them. Clear pictures will make the reader's reading experience better. Numbers taken with cheap cameras may not impress readers.
Response: Over-fit is particularly prone to occur when the data is small, so we take the residual network as one of the leading training network structures in this paper. Figure 8 shows the training results of standard deep learning networks and the comparison result of the verification accuracy rate Resnet50 has reached the highest. It also verifies the correctness of the choice. During the training process, we set 80% of the images as the training set and 20% as the validation set. This can be set automatically in the program without separating the data manually. We have added a description to the text.
The photos in Figure 5 mainly show the data's comprehensiveness and the original data's good and bad conditions. Because in the actual detection process, we cannot ensure that each detection environment is consistent and perfect. Data collected under various conditions are used for training, which benefits the training results' broad applicability. In Figure 9, Figure 10, and Figure 11, we selected specific data results for analysis.
- In Part 4, I suggest we discuss the experimental data in more detail.
Response: The data presentation in 4.1 verifies the accuracy and superiority of the semantic segmentation results. The data presentation in 4.2 verifies the effectiveness of the crack-connecting algorithm. In 4.3, we select four rail images to directly use the algorithm of this paper to detect. The test results verify that this paper can achieve similar results.
- For the fifth part, only the method in this paper is compared with magnetic particle testing. It is not compared with other methods mentioned in this article. The advantages and disadvantages of this method can be compared with those of other methods mentioned above to explain in more detail why this method has better application prospects. According to the author's experiment, the accuracy is only 94.6%, which may lead to the failure to detect the cracks on the rail surface.
Response: Following the reviewer's request, we have added a comparison of the results of other scholars using other machine learning algorithms and deep learning algorithms. And some scholars use the accurate research results of eddy current and magnetic particle testing.
|
Method |
Objtect |
Accrancy |
|
|
Ours |
Semantic Segmentation |
Defect |
96.4% |
|
Cracks |
87.83% |
||
|
Article [37] |
Bayesian classification |
Defect |
75% |
|
Article [38] |
YOLOv4 |
Defect |
92.54% |
|
Article [39] |
k-Means Clustering |
Defect |
90.9% |
|
Article [40] |
Eddy Current |
Crack Depth < 2 mm |
86% |
|
Crack Depth: 2-6mm |
88% |
||
|
Article [41] |
Magnetic Particle Inspection |
Crack Length: 2.5-5.0mm |
95% |
|
Crack Length: 2-2.5mm |
50% |
||
|
Crack Length: 1-1.5 mm |
30% |
||
|
Crack Length < 1 mm |
0% |
||
- In addition, this article is a highly specialized article, in which some abbreviations are used. For ordinary readers, these abbreviations will make it difficult for them to read the article. I suggest providing explanations where these abbreviations first appear. Moreover, the writing of this paper needs a lot of improvement in grammar, spelling and expression. This paper needs careful English polish, because there are many typing mistakes and poorly written sentences.
Response: Native English speakers have corrected Spelling errors and sentences.
The following abbreviations or specialized words have been explained in the article:
Atrous Spatial Pyramid Pooling (ASPP)
Rolling contact fatigue (RCF)
multibody simulation (MBS)
full convolutional network (FCN)
onvolutional neural networks (CNN)
magnetic particle inspection (MPI)
Deeplabv3+ is a semantic segmentation architecture that improves upon DeepLabv3 with several improvements, such as adding a simple yet effective decoder module to refine the segmentation results.
ResNet is Residual neural network.
U-net, SegNet, Xception, and Inceptionresnetv2 – which are prevalent networks in recent years.
Reviewer 3 Report
Dear Sir/Madam,
Thank you for giving the opportunity to review your manuscript. The following comments on the manuscript are given below:
(i) In most of the sections, general points will be there. Kindly avoid such points.
(ii) In the Introduction part, authors mentioned about existing technology and challenges in the current scenario. But in the Section 1.3, suddenly appears summary of contributions made. There is no link between what literature proposed and where is the gap.
(iii) In the Section 2, authors proposed the related research and methods. In this section what is different from the Introduction and subsection part.
(iv) Please clearly mention what are the methods is available in simple manner for crack detection system for the rail surface defects and also mentioned the advantages / disadvantages over the authors solution.
(v) In the results and discussion section, Table 1 gives accuracy comparison of authors proposed method. Authors must also include the other methods. Authors proposed only surface crack growth and what about depth of cracks and how will you address this issues.
(vi) How do authors ensure your method is better when compared to others based on the experience.
Thanking you and regards,

Author Response
Dear Reviewer:
We are very grateful to Reviewer for reviewing the paper so carefully. We have tried our best to improve and made some changes in the manuscript.
Responds to the reviewers' comments:
- In most of the sections, general points will be there. Kindly avoid such points.
Response: According to the reviewer's suggestion, these places have been changed to normal paragraphs, and the points have been removed.
- In the Introduction part, authors mentioned about existing technology and challenges in the current scenario. But in the Section 1.3, suddenly appears summary of contributions made. There is no link between what literature proposed and where is the gap.
Response: Section 1.2 was removed as suggested by the reviewer. We combine some of them with supplementary content as a link between the previous and the next.
- In the Section 2, authors proposed the related research and methods. In this section what is different from the Introduction and subsection part.
Response: The last part of Section 1 mainly introduces the standard defect detection techniques. We chose to use relatively efficient and cheap machine vision inspection as the basis. We tried to propose a technology based on machine vision inspection to achieve the effect of magnetic particle inspection.
The latter part mainly introduces several popular algorithms in developing machine vision detection technology. This part especially gives the reader a brief understanding of machine vision algorithms. Moreover, found the residual network algorithm suitable for this dataset is small and prone to Over-fit.
- Please clearly mention what are the methods is available in simple manner for crack detection system for the rail surface defects and also mentioned the advantages / disadvantages over the authors solution.
Response: According to the reviewer's suggestion, we supplement the rating results of Foerste GmbH at the end of this Section 1. This company rates several primary detection methods. The scoring results vary widely in the face of different situations. The advantages of each method can be significant compared through the rating results.
|
Flaw Type Inspection Method |
Eddy Current |
X-Ray |
Magnetic Particle |
Ultrasonic |
|
|
Surface Breaking, Linear |
A |
C |
A |
D |
|
|
Surface Breaking, Linear & Parallel to Surface |
A |
D |
A |
D |
|
|
Near-Surface |
B |
A |
B |
B |
|
|
Subsurface, Volumetric |
D |
A |
D |
A |
|
|
A = Best suitable; B = Kind of OK to use; C = Not so good; D = Not at all suitable |
|||||
- In the results and discussion section, Table 1 gives accuracy comparison of authors proposed method. Authors must also include the other methods. Authors proposed only surface crack growth and what about depth of cracks and how will you address this issues.
Response: Following the reviewer's request, we have added a comparison of the results of other scholars using other machine learning algorithms and deep learning algorithms. And some scholars use the accurate research results of eddy current and magnetic particle testing.
|
Method |
Objtect |
Accrancy |
|
|
Ours |
Semantic Segmentation |
Defect |
96.4% |
|
Cracks |
87.83% |
||
|
Article [37] |
Bayesian classification |
Defect |
75% |
|
Article [38] |
YOLOv4 |
Defect |
92.54% |
|
Article [39] |
k-Means Clustering |
Defect |
90.9% |
|
Article [40] |
Eddy Current |
Crack Depth < 2 mm |
86% |
|
Crack Depth: 2-6mm |
88% |
||
|
Article [41] |
Magnetic Particle Inspection |
Crack Length: 2.5-5.0mm |
95% |
|
Crack Length: 2-2.5mm |
50% |
||
|
Crack Length: 1-1.5 mm |
30% |
||
|
Crack Length < 1 mm |
0% |
||
- How do authors ensure your method is better when compared to others based on the experience.
Response: The primary purpose of this paper is not to find a more accurate detection method. Instead, we are looking for a technique that can achieve a similar magnetic particle detection effect in rail surface damage detection. We need to achieve low-cost and fast results without losing the primary information about the defect. Moreover, machine vision-based detection does not require detection workers to have long-term monitoring experience and training. Because of today's detection requirements and higher frequency of faults, the method proposed in this paper promises to meet the above needs while reducing costs.
Round 2
Reviewer 2 Report
It can be accepted in present form
Author Response
Thanks to the reviewer for your appreciation, we have made small changes to the grammar of the article as you requested.